# Role of Cop9 Signalosome Subunits in the Environmental and Hormonal Balance of Plant

**DOI:** 10.3390/biom9060224

**Published:** 2019-06-09

**Authors:** Amit Kumar Singh, Daniel A. Chamovitz

**Affiliations:** 1School of Plant Sciences and Food Security, Tel Aviv University, Tel Aviv 6997801, Israel; dannyc@tauex.tau.ac.il; 2Jacob Blaustein Institutes for Desert Research, Ben-Gurion University of the Negev, Midreshet Ben-Gurion 8499000, Israel

**Keywords:** COP9 signalosome, CSN subunit, hypomorphic mutants, biotic stress, abiotic stress, hormonal signaling

## Abstract

The COP9 (Constitutive photomorphogenesis 9) signalosome (CSN) is a highly conserved protein complex that influences several signaling and developmental processes. The COP9 signalosome consists of eight subunits, among which two subunits, CSN5 and CSN6, contain an Mpr1/Pad1 N-terminal (MPN) domain and the remaining six subunits contain a proteasome, COP9 signalosome, and initiation factor 3 (PCI) domain. In plants, each MPN subunit is encoded by two genes, which is not the case in other organisms. This review aims to provide in-depth knowledge of each COP9 signalosome subunit, concentrating on genetic analysis of both partial and complete loss-of-function mutants. At the beginning of this review, the role of COP9 signalosome in the hormonal signaling and defense is discussed, whereas later sections deal in detail with the available partial loss-of-function, hypomorphic mutants of each subunit. All available hypomorphic mutants are compared based on their growth response and deneddylation activity.

## 1. Introduction

The COP9 (Constitutive photomorphogenesis 9) signalosome (CSN) is an eight-subunit protein complex which is conserved throughout evolution. It was first identified in Arabidopsis as a repressor of light-regulated development [1,2]. Regulation of protein degradation by deneddylation of cullin-RING E3 ligase (CRL) is the most studied biochemical function of CSN. CSN subunits are ordered from CSN1 to CSN8 on the basis of their descending molecular mass. Two of the subunits, CSN5 and CSN6, possess an Mpr1/Pad1 N-terminal (MPN) domain and the remaining six contain a proteasome, COP9 signalosome, and initiation factor 3 (PCI) domain. Both MPN and PCI domains are also present in lid subunit of 26S proteasome and eukaryotic translation initiation factor 3 (eIF3) [3].

Viable CSN mutants with reduced activity demonstrate that CSN plays a major role in the growth and development of the plant. RNA silencing of CSN5 subunit in Arabidopsis reduces auxin signaling [4]. Silencing of CSN3 and CSN6 subunits reveal the role of CSN in floral development. The various floral phenotypes (short stamens, less pollen grains and reduced number of petals) in these strains are suppressed by overexpression of the F-box protein UFO (unusual floral organs) hence, CSN is also linked with SKP1, Cullin and F-box-containing protein (SCF)^UFO^ [5]. Further, it was found that Floral B domain transcription factor APETALA3 is down-regulated in strains with compromised CSN1 function [6]. CSN also regulates defense against pathogens by playing a crucial role in *N* gene-mediated resistance to tobacco mosaic virus [7], and in jasmonic acid-dependent plant defense responses [8]. Studies in human cells showed that the CSN also plays a vital role in repair of double-strand breaks by getting recruited to double-strand break sites in a neddylation-dependent manner [9], and knockdown of CSN results in a defect of nucleotide excision repair [10]. In Arabidopsis, CSN also plays an important role in the DNA repair mechanism. *csn7* null mutants are tolerant to ultraviolet (UV)-C treatment due to constitutive expression of UV-induced genes as CSN negatively regulates ribonucleotide reductase activity [11]. Thus, the role of CSN in plants is not only limited to hormonal signaling and defense response but also has a critical impact in many cellular, developmental processes [12] as well as in cell cycle progression [13].

## 2. Role of CSN in Hormonal Signaling

The CSN plays a central role in hormonal signaling in plants, with the auxin-mediated response being a highly studied paradigm. The interaction of CSN with CRL E3 ubiquitin ligase was first identified in the study that focused on the role of CSN in SKP1, Cullin and F-box-containing protein- Transport Inhibitor response (SCF^TIR1^) mediated auxin response [4]. In response to auxin perception, SCF^TIR1^ mediates degradation of the auxin/3-indoleacetic acid (AUX/IAA) transcriptional repressors. A transgenic version of the *Pisum*-AUX/IAA protein IAA6 introduced in Arabidopsis, was partially stabilized in a transgenic Arabidopsis strain expressing a CSN5 antisense construct. This partial stabilization of psIAA6 impaired auxin signaling in these plants [4]. In addition to interacting with SCF^TIR1^, the CSN also interacts with other CRLs such as the SCF specific for jasmonic acid signaling, SCF^COI1^, gibberellic acid signaling, SCF^SLY1^, flower development, SCF^UFO^, as well as several other E3 ligases [6,14,15].

Similar to AUX/IAA protein stabilization in the absence of auxin, it was also found that DELLA proteins (repressor of gibberellic acid signaling) accumulate at a higher amount in the absence of gibberellic acid (GA). DELLA proteins interact with phytochrome interacting factor 3 (PIF3) and prevent the binding of PIF3 to the target gene promoters. In the presence of GA, DELLA proteins are degraded releasing the PIF3 from the negative suppression of DELLA protein [16]. The significance of CSN in ethylene signaling was shown through the analysis of the Arabidopsis *eer5-1* mutant. These mutants have enhanced ethylene response in etiolated seedlings and are hypersensitive to ethylene. *eer5-1* is formed by amino acid substitution in a PCI-Associated Module (PAM). Enhanced ethylene response protein 5 (EER5) interacts with C-terminal of ethylene-insensitive protein 2 (EIN2) and CSN, forming a bridge between EIN2 and transcriptional repressor during ethylene signaling [17]. CSN also regulates seed germination by promoting degradation of RGL2 and ABI5 protein [18]. RGL2 is GA signaling repressor whereas ABI5 is an effector of ABA response. *csn* mutants are impaired in the timely removal of RGL2 and ABI5. Arabidopsis contains five DELLA proteins (RGL1, RGL2, RGL3, GAI and RGA) which negatively regulates GA pathway; among these RGA and GAI functions mostly in dark to inhibit seed germination whereas, RGL2 works in light [19]. Seed germination inhibition by RGL2 is seed coat-dependent which probably involves CSN. RGL2 is regulated by SCF^SLY1^ E3 ligase-mediated protein degradation through proteasome [20].

## 3. Importance of CSN in Biotic and Abiotic Stress

Together with hormonal signaling, the CSN also plays a crucial role in plant defense response. The role of CSN in plant defense becomes evident during the studies of tomato defense response using tobacco rattle virus (TRV) based virus-induced gene silencing (VIGS). Expression of CSN5 was reduced using VIGS, and as a result CSN5-VIGS plants were 50% stunted compared to control VIGS plants [8]. Reduced synthesis of jasmonic acid (JA) in CSN5-VIGS plants result in poor defense against *Manduca Sexta* and *Botrytis cinerea*. This shows the importance of CSN and SCF^COI1^ interaction, which is not only essential for plant development but also for plant defense. Although CSN5-VIGS plants show reduced resistance to herbivorous *M. sexta* and necrotrophic fungus *B. cinerea*, its susceptibility to tobacco mosaic virus (TMV) remains unaltered. Study of gene expression profiles revealed that JA-dependent wound genes are positively regulated by CSN, while salicylic acid (SA)-responsive pathogenesis related (PR) genes are negatively regulated by CSN. PR genes are constitutively high in CSN-VIGS plants which make it less susceptible to TMV. This study showed that CSN is important for JA synthesis but not for the synthesis of SA [8]. Another study showed that gemini viral C2 protein compromised CSN activity on CUL1. C2 protein only affects CUL1 but not other cullins (CUL3 and CUL4) and this fact was confirmed by studying levels of neddylated cullin. The transgenic line containing C2 protein showed a higher level of neddylated CUL1 whereas, levels of CUL3 and CUL4 neddylation remains unaffected. Further, C2 protein interacts with CSN5 and alters several hormonal pathways (including auxin, JA, GA, ethylene, and ABA) which are regulated by CUL1-based SCF ubiquitin E3 ligases. Stabilization of GAI (substrate of SCF^SLY1^) in the transgene containing C2 protein validates the impairment of SCF function. Transcriptomic data of this transgenic plant reveals that jasmonate response is most severely affected due to the impairment of SCF function by C2. Gemini virus infection gets diminished by exogenous application of JA indicating that inhibition of jasmonate response is important for infection [21].

Plants perceive biotic and abiotic signals differently. Biotic signals are receptor-mediated, whereas abiotic signals are mostly sensed by perturbation of the plasma membrane [22]. Plants utilize numerous hormonal signaling pathways in coping with abiotic stress. ABA and ethylene are the most frequent stimulant of abiotic stress which cross-talk with SA, JA, and auxin to change the gene expression profile for adaptive plant response against abiotic stress [23,24]. The role of CSN in abiotic stress is further supported by the study of differentially expressed proteins in heat tolerant and heat sensitive lines of rice. With the help of iTRAQ LC-MS/MS, it was found that 38 proteins are differentially expressed between heat tolerant and heat sensitive lines of rice. Among these, calcium-dependent protein kinase, COP9 signalosome, and bHLH transcription factor were up-regulated in heat tolerant lines, suggesting further effects downstream during high night temperature response [25].

## 4. Characteristics of CSN Hypomorphic Mutants and Their Importance to Study Plant Growth and Development

Early seedling-lethality of *csn* null mutant hinders the proper analysis of CSN in physiological and developmental functions in Arabidopsis [26]. Discovery of viable CSN hypomorphic mutants that can survive to the adult stage as well as reproduce, paved the way for in-depth analysis of CSN’s role in the developmental and physiological response of the mature plants. Reduction in function of different CSN subunits leads to different phenotypes which may indicate that each CSN subunits has unique roles [4,5,27]. Thus, CSN’s function can be regulated by engineering particular subunits [28]. The following sections review data available for the reported hypomorphic mutants of CSN subunits.

### 4.1. csn1-10

*csn1-10* is a weak hypomorphic mutant formed by a transition mutation converting Ser-305 into Asn. Ser at this position is highly conserved among CSN1 orthologs. This mutation is in the last base of exon 4, and thus it affects mRNA splicing and results in the formation of two mRNA species. One mRNA is formed by correct splicing while the second is produced by short deletion, resulting in a frameshift that truncates the protein due to splicing at site 32 bases upstream of the normal splice site [29]. Western blot analysis confirms a reduced level of full-length CSN1 protein in this hypomorphic mutant. As opposed to previously isolated loss-of-function *csn1* mutants [29], *csn1-10* hypomorphic mutants are viable and do exhibit *cop* phenotype. *csn1-10* seedlings show auxin-resistant root growth and diminished SCF^TIR1^ activity. Thus, *csn1-10* is a weak allele which contains adequate CSN function for viability and suppression of photomorphogenesis, though the deneddylation activity against CUL1 is partially compromised [29].

### 4.2. csn2-5

*csn2-5* is formed by the exchange of a single base pair causing amino acid substitution of Gly-237 into Asp. Like other *csn* mutants, *csn2-5* mutant is auxin-resistant, and CUL1 is constitutively neddylated [30]. However, the level of CUL1 neddylation is less than in the null *csn2* allele. The total amount of CUL1 and CUL3 is decreased significantly in *csn2-5* hypomorphic mutants compared to the wild type. However, *csn2-5* mutant does not show obvious developmental phenotypes other than slight dwarfism. In darkness, relative hypocotyl elongation is similar to wild type and does not have apparent *cop* phenotype. Thus, *csn2-5* is a viable fertile hypomorphic mutant. This suggests that one should not expect an exact correlation between CUL modification (neddylation/abundance) and developmental response.

### 4.3. csn3-3

*csn3-3* was found during the screening for enhancer mutants of the auxin response defect in the tir1-1. One such mutant, *eta*, contains single point mutation in the PCI domain of CSN3 [31] converting Gly-293 into Glu. Like *csn1-10*, *csn3-3* also shows auxin-resistant root growth. But unlike *csn1-10*, it rapidly degrades AUX/IAA which is confirmed with the help of IAA28-myc and HS:AXR3NT-GUS constructs. Thus, the activity of SCF^TIR1/AFB^ remains unchanged in the *csn3-3* mutation. Also, *csn3-3* does not exhibit the *cop* phenotype. Further, unlike other *csn* mutants, *csn3-3* does not affect deneddylation activity of CUL1 and CUL4, nor is the CSN holocomplex assembly affected. *csn3-3* confers auxin resistance without influencing cullin deneddylation, indicating that CSN3 have another role in auxin signaling together with the role of regulating SCF^TIR1/AFB^ ubiquitin ligase [31].

### 4.4. csn4-2035

*csn4-2035* is formed by G to A mutation at position 2592 in the tenth exon of CSN4 gene, resulting Ala-302 into Val substitution [32]. Ala-302 is highly conserved and part of the helix-loop-helix of the PCI domain that acts as scaffold for CSN4-6-7 interaction in Arabidopsis [33]. *csn4-2035* roots are resistant to 2,4-Dichlorophenoxyacetic acid. *csn4-2035* is inefficient in AUX/IAA protein degradation and also shows accumulation of neddylated CUL1 similar to *csn1-10*, indicating that *csn4-2035* is partially defective in CSN-dependent deneddylation activity.

### 4.5. csn5a-1, csn5a-2 and csn5b-1

In Arabidopsis, CSN5 is encoded by two genes *CSN5A* and *CSN5B* [34]. T-DNA insertion lines in CSN5A and CSN5B produces *csn5a-1* (T-DNA inserted in the exon of At1g22920), *csn5a-2* (T-DNA inserted in the intron of At1g22920) and *csn5b-1* (T-DNA inserted in Atg71230) [35]. *csn5a-1* and *csn5a-2* exhibit reduction in growth, lateral root formation, root hairs formation and flower size. Yet both *csn5a* mutants are fertile and propagate as homozygous mutants. *csn5a* mutants (*csn5a-1* and *csn5a-2*) are insensitive to auxin-mediated root growth inhibition as well as auxin stimulated GH3-2: LUC. In the initial days of seedling growth, *csn5a-1* and *csn5a-2* are indistinguishable from each other, while after 8 days the growth defects of *csn5a-1* become more pronounced, especially in the case of rosette size and trichome density. While single mutants for *csn5a* (*csn5a-1* or *csn5a-2*) are partially blocked in auxin stimulated expression of GH3-2: LUC reporter, a *csn5a-2*; *csn5b-1* double mutant is fully blocked in auxin stimulated expression of GH3-2: LUC reporter. Moreover, these double mutants have *cop* phenotype, characterized by accumulation of anthocyanin, short hypocotyl, open cotyledon and expression of light induced genes in dark grown seedlings. On the other hand, the *csn5b-1* mutant does not show any detectable growth abnormality and is essentially indistinguishable from the wild type [12]. While *csn5a* mutants show significant accumulation of neddylated CUL1, *csn5b-1* does not have any defect in cullin neddylation [35]. Thus, CSN5A, not CSN5B play a major role in plant development [12].

### 4.6. csn6a-1 and csn6b-1

CSN6 subunit is encoded by two genes CSN6A and CSN6B. The protein encoded by these two genes show 87% identity. T-DNA insertions in these two genes produce null mutants named *csn6a-1* and *csn6b-1*. *csn6* mutants exhibit a slight reduction in hypocotyl length in dark and blue light compared to the wild type. *csn6a-1* exhibit slightly more effect on hypocotyl elongation compared to *csn6b-1* in dark but both mutants do not show any morphological defect in white light. Lack of *cop* type phenotype in single *csn6* mutants showed that both genes act in a redundant manner in regulating photomorphogenesis [12]. *csn6a* mutants do not affect the stability of other CSN subunits but *csn5a* mutants change the cellular pool of CSN subunit dramatically. However, complete loss of CSN6 (*csn6a-1 csn6b-1*) reduced the protein levels of CSN1, CSN3, CSN4, CSN7 and CSN8. CSN2 and CSN5 are not affected by the loss of CSN6. CSN2 and CSN4 are affected differently in various *csn* null mutants hence they should not be used to study CSN depletion. Cellular levels of CUL1, CUL3 and CUL4 also get affected differentially in CSN hypomorphic mutants. CUL1 remains unchanged as it is just redistributed from modified to unmodified form in PCI or MPN loss of subunits. CUL3 is significantly reduced in CSN hypomorphic mutants further; RUB-CUL3 is hardly detectable in *csn* null mutants. However, the total cellular pool of CUL4 increased significantly in CSN mutants compared to wild type. CSN5 and CUL3 regulate each other reciprocally in such a manner that loss of either CUL3A or CUL3B results in higher accumulation of CSN5 protein and mutation in CUL3A or CUL3B suppress pleiotropic phenotype of *csn5a* mutant.

## 5. Comparison of Hypomorphic Mutants on the Basis of Developmental and Deneddylation Activity

The viable hypomorphic mutants of CSN provided an opportunity to understand the role of an individual COP9 subunit beyond the seedling stage. Although most of the CSN hypomorphic mutants show similar phenotypes such as reduction in growth and impaired hormonal activity, their level of developmental defects does not remain the same. *csn5a* mutants display several photomorphogenic and developmental defects during seedling and adult stage such as reduction in lateral root and root hair formation but *csn5b* mutants show only mild phenotype. Nevertheless, double mutants of *csn5a-1 csn5b* and *csn5a-2 csn5b* show the similar phenotype of *cop/det/fus* demonstrating that two *CSN5* genes have redundant function with a stronger contribution of *CSN5A*. Contrarily, with the exception of mild photomorphogenic phenotype in dark and in blue light, *csn6a* and *csn6b* mutants do not show any phenotypic defect in white light. However, loss of function of both genes (*csn6a-1 csn6b-1* double mutant) leads to seedling lethality and *cop/det/fus* phenotype. Among the three PCI subunit viable mutants (*csn1-10*, *csn2-5*, and *csn3-3*), *csn1-10* show severe pleiotropic developmental defect together with impaired auxin response, *csn2-5* displays mild dwarfism with curly hypocotyl when grown in dark whereas, *csn3-3* exhibit wild type phenotype although it is impaired in auxin response.

Adventitious and lateral root formation differed among the various mutants. *csn5a-1* and *csn5a-2* do not form any adventitious roots, while *csn5b* formed a higher number of adventitious roots. Similar to the *csn5a* mutants, *csn1-10* and *csn3-3* develop very minimal adventitious roots. *csn2-5* produced adventitious roots similar to its wild type. Most of the *csn* mutants (*csn1-10*, *csn3-3*, *csn5a-1*, and *csn5a-2*) suppress adventitious root phenotype of *sur2-1* (*superroot 2-1*) other than *csn5b* which enhances *sur2-1* phenotype. Thus, the two subunits of CSN5 (CSN5A and CSN5B) regulate adventitious root formation differentially. *csn4-2035* mutant show a different impact on adventitious and lateral root development [32]. *csn4-2035* suppresses adventitious root phenotype of *sur2-1* potentially by affecting auxin-jasmonate-light cross talk.

Regarding lateral root formation, all *csn* mutants (*csn1-10, csn2-5*, *csn3-3*, *csn5a-1, csn5a-2*, and *csn5b*) showed a decreased lateral root formation in the seedling stage. Interestingly, although *csn2-5* shows decrease in the number of lateral root formation, it does not show any change in adventitious root formation. Hence, CSN subunits have a different role in adventitious and lateral root development.

CSN subunits also show differential regulation in seed germination [18]. *csn1-10*, *csn3-3*, *csn5a-1,* and *csn5a-2* exhibited poor seed germination with a distinct delay in comparison to Col. *csn2-5* exhibit mild but consistent poor seed germination in comparison to wild type Ler. Germination defects can be completely overcome by cold stratification in *csn1-10*, *csn3-3*, and *csn2-5* seeds, but not in *csn5a-1* and *csn5a-2*. Thus, most of the *csn* mutants show strong seed dormancy except *csn5b-1* which show weaker dormancy than Col. *csn5b-1* germinates well regardless of whether the seed has been cold stratified or not.

CSN regulates seed germination by degrading RGL2 (a repressor of GA signaling) and ABI5 (an effector of ABA signaling) [18]. Although *csn1-10* and *csn5a-1* display similar seed germination problem, there are many differences between them. First, *csn1-10* exhibit deeper seed dormancy which can be restored by cold stratification or by prolonged after-ripening period but *csn5a-1* additionally shows delayed germination which cannot be rescued by these dormancy breaking methods. Second, loss of RGL2 can suppress germination phenotype of *csn1-10* but not in *csn5a-1*. Third, ABA synthesis inhibitor norflurazon can rescue the germination defect of *csn1-10* but not of *csn5a-1*. Fourth, transcriptome profile of *csn1-10* and *csn5a-1* varies at different developmental stages. Fifth, *csn1-10* only affects seed germination while *csn5a-1* affects both seed germination as well as seed formation. Sixth, the germination phenotype of *csn1-10* is caused by over-accumulation of RGL2, whereas the germination phenotype of *csn5a-1* is caused by over-accumulation of both RGL2 and ABI5 [18]. The observation that ABI5 is especially affected in *csn5a* but not in *csn1-10* shows the first *CSN5* specific activity in the plant. Transcriptome data exhibited a prominent role of *CSN1* in seed maturation compared to *CSN5A* however; *CSN5A* had a major role in seed germination. This finding again supports the different role of CSN subunits.

Null mutants of each CSN subunit are defective in cullin neddylation. But this is not true with all CSN hypomorphic mutants. Hypomorphic mutants such as *csn5a-1*, *csn5a-2*, *csn1-10*, and *csn2-5* are defective in deneddylation, but *csn5b-1*, *csn3-3*, *csn6a-1,* and *csn6b-1* do not show any defect in cullin deneddylation. Deneddylation activity or reduction of CSN subunits are not the only criteria to judge the severity of CSN mutant, considering that *csn2-5* and VIGS of CSN5 show severe impairment in cullin neddylation in their respective subunits (50% in *csn2-5* and 90% in VIGS of CSN5) but display mild dwarfism. Also, the hormonal responses of CSN subunit mutants are not similar, for example; *csn2-5* mutants are impaired in auxin signaling but response normally to JA and ethylene. However, CSN5-VIGS tomato plants are especially impaired in JA biosynthesis.

CSN mutants also show difference in their ploidy content. Flow cytometry analysis in root tips of *csn* null mutants such as *csn3*, *csn4* and *csn5ab* show delay in G2 phase with a higher 4C DNA content of 65–70% compared to 28–34% in wild type. 4C content of hypomorphic mutants show different percentage (*csn5a-1* 62%, *csn5a-2* 52% and *csn5b-1* 29.5%) [13]. Therefore, hypomorphic mutants of CSN subunits again show variation and this time in the form of ploidy.

## 6. Conclusions and Future Perspective

This review provides comprehensive information about the role of CSN in physiological, developmental and defense mechanisms. The availability of hypomorphic mutants offers an opportunity to elucidate the specific role of each subunit beyond the seedling stage (summarized in Table 1).

The role of CSN in protein degradation is a crucial step for the maintenance of proper plant growth. CSN performs this work by deneddylation of cullin. The interaction of CRL-CSN is not a simple process because of a high number of CRL substrate and substrate adaptors in plants. Dimerization of CRLs further enhance this complexity. Hence, neddylation together with dimerization adds an extra regulatory mechanism for CRLs to accommodate different substrates. For example, SCF^CDC4^ dimerization enhances the rate of ubiquitination but does not affect substrate recruitment. It is known that deneddylation of cullin by CSN is absolutely essential for CRL functioning but null mutants of CSN and cullin does not lead to similar developmental defect. *csn* null mutants result in post-embryonic lethality during seedling stage whereas, cullin mutants (*cul3* and *cul4)* results in embryonic arrest at the globular stage. This difference could be either due to the presence of residual deneddylation activity from maternally contributed CSN or CSN may not be absolutely essential for cell division at least during embryogenesis. This view is further supported by an experiment which shows the seedling growth arrest of *csn* mutant is associated with G2 phase delay not due to arrest of cell division because the cell division still happens in the roots of *csn* mutant. Finding the molecular mechanism of this phenomenon can increase our understanding of CSN during embryogenesis.

CSN hypomorphic mutants provided an important tool to understand the role of each CSN subunit in plant development, which could not have been possible with *csn* null mutants. Hypomorphic mutants of CSN individual subunits display a distinct range of abnormalities. For instance, *csn2-5* mutants are resistant to auxin due to lower FBP TIR1 stability, but this is not the case with *csn3-3* mutants. Most of the CSN hypomorphic mutants which are resistant to auxin display dwarf phenotype but *csn2-5* although being defective in auxin response does not show any prominent developmental phenotype. Phenotypic differences of CSN hypomorphic mutants on the basis of deneddylation, AUX/IAA degradation and auxin response have been represented in (Figure 1).

However, these comparative analyses must be tempered by the possibility that the differences noted are not so much subunit specific as they are related to the relative severity of the mutations. Thus, the availability of additional hypomorphic mutants for each subunit will be essential for elucidating subunit-specific roles of the CSN.

## Figures and Tables

**Figure 1 biomolecules-09-00224-f001:**
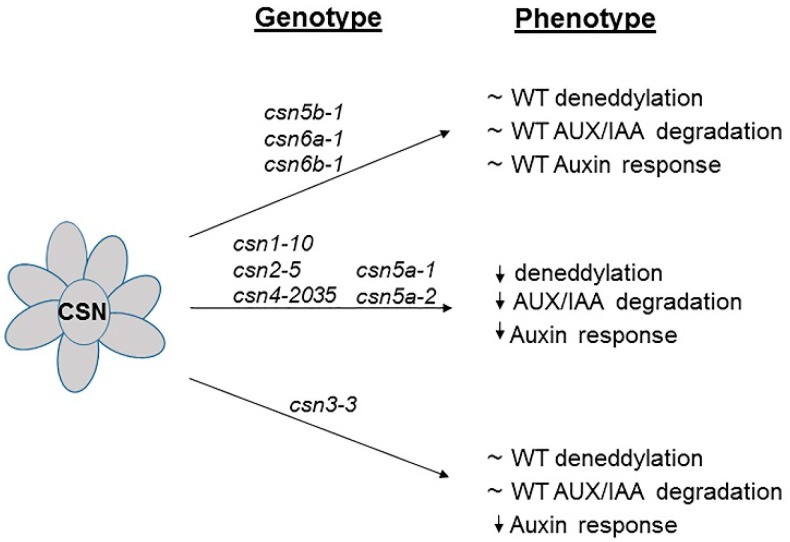
Schematic representation of the influence of hypomorphic mutants of the CSN on deneddylation, AUX/IAA degradation and auxin response.

**Table 1 biomolecules-09-00224-t001:** Characterization of CSN hypomorphic mutants.

Hypomorphic Mutants	Ecotype	Details of Mutation and Phenotype	Deneddylation Activity	References
*csn1-10*	Col-0	Transition mutation resulting in Ser-Asn substitution at amino acid 305. Mild dwarf phenotype. Show auxin-resistant root growth and reduced SCF^TIR1^ activity.	Affects	[29]
*csn* *2-5*	Ler	Single base pair exchange in codon GGT-GAT results in amino acid substitution mutation from Gly-Asp at position 237 in CSN2. Hypomorphic mutant, viable and fertilize. Mild dwarf phenotype. Show auxin-resistant root growth and defective in Aux/IAA protein degradation.	Affects	[30]
*csn* *3-3*	Col-0	Missense mutation in Gly-293 to Glu of PCI domain of CSN3. Exhibit auxin related response like auxin-resistant root growth. But unlike *csn* mutants, it is not defective in Aux/IAA protein degradation.	Does not affect	[31]
*csn4-2035*	Ws-4	Point mutation of G to A at position 2592 in the tenth exon of CSN4 results in Ala-302 to Val amino acid substitution. Slightly resistant to exogenously applied auxin. Inefficient degradation of AUX/IAA protein. Produced similar amount of CSN4 protein as the wild type. Dwarf phenotype with reduction in the number of adventitious root formation.	Partially affects	[32]
*csn* *5a-1*	Col-0	T-DNA insertion (SALK_063436). Severe dwarf phenotype. Auxin-resistant root growth.	Affects	[35]
*csn5a-2*	Col-0	T-DNA insertion (SALK_027705). Dwarf phenotype. Affects deneddylation activity. Auxin and jasmonic acid resistant root growth.	Affects	[35]
*csn* *5b-1*	Col-0	T-DNA insertion (SALK_007134). Phenotype almost similar to wild type (Col-0). Auxin and jasmonic acid sensitive root growth.	Does not affect	[35]
*csn6a-1* and *csn6b-1*	Col-0	T-DNA insertion SALK_146926 (*csn6a-1*), and SALK_036965 (*csn6b-1*), In the dark, loss of CSN6A affect hypocotyl elongation slightly more than loss of CSN6B. But both the mutants do not display any obvious morphological defects in white light.	Do not affect	[12]

SCF: SKP1, Cullin and F-box-containing protein, TIR1: Transport Inhibitor response, Aux/IAA: auxin/3-indoleacetic acid protein.

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
