# Peer review of "Role of Cop9 Signalosome Subunits in the Environmental and Hormonal Balance of Plant"

_biomolecules, 2019, doi:10.3390/biom9060224_

Round 1
Reviewer 1 Report
In this review, Singh and Chamovitz summarizes literature on plant COP9 signalosome complex. They describe mutants of each subunit and their role in environmental interaction and hormone balance in terms of plant growth. Dr. Chamovitz made groundbreaking discoveries on COP9 complex and is an expert on them.
Here are some minor points-
I felt this review is a bit wordy. A cartoon depiction of the COP9 signalosome complex and its subunits would definitely improve the visual appeal and readability.
In the abstract Proteasome is mentioned as a domain? Is that true?
Line 222, "later" should read lateral.
In line 258, this sentence is not making much sense.
Author Response
Answer to reviewer
In this review, Singh and Chamovitz summarizes literature on plant COP9 signalosome complex. They describe mutants of each subunit and their role in environmental interaction and hormone balance in terms of plant growth. Dr. Chamovitz made groundbreaking discoveries on COP9 complex and is an expert on them.
Here are some minor points-
1. I felt this review is a bit wordy. A cartoon depiction of the COP9 signalosome complex and its subunits would definitely improve the visual appeal and readability.
Answer: A figure showing the Schematic representation of the influence of hypomorphic mutants of the CSN on deneddylation, AUX/IAA degradation and auxin response has been integrated in the manuscript now.
2. In the abstract Proteasome is mentioned as a domain? Is that true?
Answer: The word "domain" is use for PCI (Proteasome, COP9 signalosome, and initiation factor 3) domain of CSN subunits in the abstract.
3. Line 222, "later" should read lateral.
Answer: The word "later" has been replaced with "lateral" (line 222).
4. In line 258, this sentence is not making much sense.
Answer: This sentence has been deleted (line 258).
Reviewer 2 Report
The COP9 signalosome (CSN) is a multiprotein complex highly conserved in all Eukaryotes which controls many regulatory pathways involved in response to the environment, cell division, hormone responses etc.... Loss of function mutations in any of the 8 subunits is lethal which has hampered so far to dissect the specific roles of the CSN subunits. The recent identification of partial loss of function mutants in plants opens new possibilities for a better understanding of the CSN functioning.
The present manuscript is a short and straightforward review about the COP9 signalosome in plants, its different subunits, their role in the complex, and how the recent identification of partial loss of function mutants could help dissecting the specificity of action of the CSN in the different regulatory pathways which it controls.
The manuscript is well written and summarizes the current knowledge. A schematic representation of the complex and the organization of the subunits could helpful.
Author Response
Answer to reviewer
Comments and Suggestions for Authors
The COP9 signalosome (CSN) is a multiprotein complex highly conserved in all Eukaryotes which controls many regulatory pathways involved in response to the environment, cell division, hormone responses etc.... Loss of function mutations in any of the 8 subunits is lethal which has hampered so far to dissect the specific roles of the CSN subunits. The recent identification of partial loss of function mutants in plants opens new possibilities for a better understanding of the CSN functioning.
The present manuscript is a short and straightforward review about the COP9 signalosome in plants, its different subunits, their role in the complex, and how the recent identification of partial loss of function mutants could help dissecting the specificity of action of the CSN in the different regulatory pathways which it controls.
The manuscript is well written and summarizes the current knowledge.
1. A schematic representation of the complex and the organization of the subunits could helpful.
Answer: A figure showing the Schematic representation of the influence of hypomorphic mutants of the CSN on deneddylation, AUX/IAA degradation and auxin response has been integrated in the manuscript now.
Reviewer 3 Report
The authors give here a nice overview of the function of single CSN subunits in different steps of plant development.
For readers not coming from the COP9 signalosome or plant hormone research field it would be nice to have one or two small schemes that illustrate the description of the authors.
In general, the authors should go through the whole article to correct typos, commas, prepositions and pay some more attention to singular and plural form of words (some examples are given below).
Line 9/10: COP9 signalosome or COP9 Signalosome, uniform way of writing in the whole article
Line13: replace "..not the case with other organisms" to "..not the case in other organisms.."
Line13/14: replace "This review is aimed to provide..." with "This review aims to provide..."
Line17: replace "about" with "with" and remove comma after hypomorphic
Line18: replace "mutats" by "mutants
Line35: replace "phonotypes" by "phenotypes"
Line36: mention that UFO is an Fbox protein for readers who do not directly come from the field
Line38: replace "under-expressed" with "downregulated"
Line43: is "recruiting" the right word to use? what proteins does CSN recruit to the DNA? Or is CSN just been recruited
Line 43: replace "repairs" by "repair"
Line 43-45: "In Arabidopsis CSN...UV-induced gene expression", the meaning of the sentence is not clear
Line 57: replace "CSN not only..." with "CSN does not only..."
Line 63-66: sentence is not clear
Line 70: C-terminal
Line 86: write full names of M. sexta and B.cinerea as they are mentioned the first time here
Line 89: replace "Botrytis cinerea" by "B.cinerea"
Line 96: be consistened with the words rubylated and neddylated, replace rubylated with neddylated
Line 113: Where all subunits of the COP9 signalosome identified in this study or just some?
Line126/127: increase the spacing between the lines
Line 150: If I understood it right the csn3-3 mutant has a single amino acid exchange from glycine to glutamate. Csn3 can maybe still exert its function similar to the wild type Csn3 and therefore there is also no deneddylation phenotype. Maybe one should be a bit careful with the conclusion that Csn3 has other roles (Line158).
Line 161: change to "...position 2592..:"
Line 164: 2,4D, maybe the authors can write the full name
Line 185: space is missing
Line 186/187: Change the sentence to: "The proteins encoded by these two genes show 87% identity."
Line 194,195: what about Csn5? Even if there was no effect observed the authors can maybe mention that as this would be also interesting for the reader.
Line 195: give informationabout what is reduced like "...reduces the proteins levels of CSN1,..."
Line 198: Change to "CUL3 is significantly reduced ..:"
Line 201: write "...CUL3 regulate each other reciprocally in such a manner/way that loss of ..:"
Line 217: "..of both the genes.." replace by "..of both genes"
Line 217: Authors write that the double deletion of csn6a-1 and csn6b-1 leads to seedling lethality, but in lines 194/195 they state that in the double deletion levels of other CSN subunits are reduced? How does this fit together?
Lines 222-226: many repetitions of "adventitious", maybe the authors can replace some of these words
Line 228: replace "differently" with "differentially"
Line 239: replace "stratification" with "stress"
Line 255/256: CSN5/1/5A should not be italic
Line 279: Table1: in den column of deneddylation activity: either "affects" or "does not affect" if the authors want to describe the deneddylation activity with a verb, or "effect" or "no effect" when authors want to state the activity with a noun
Line 282: replace "CSN perform" by "CSN performs"
Line 284: replace "CRLS" by "CRLs"
Line 297: replace "CSN subunits" by "CSN subunit"
Author Response
Answer to the reviewer
The authors give here a nice overview of the function of single CSN subunits in different steps of plant development.
1. For readers not coming from the COP9 signalosome or plant hormone research field it would be nice to have one or two small schemes that illustrate the description of the authors.
Answer: A figure showing the Schematic representation of the influence of hypomorphic mutants of the CSN on deneddylation, AUX/IAA degradation and auxin response has been integrated in the manuscript now.
2. In general, the authors should go through the whole article to correct typos, commas, prepositions and pay some more attention to singular and plural form of words (some examples are given below).
Answer: The manuscript has been corrected properly and the changes have been made in track change mode.
3. Line 9/10: COP9 signalosome or COP9 Signalosome, uniform way of writing in the whole article
Answer: COP signalosome is made uniform in the article.
4. Line13: replace "..not the case with other organisms" to "..not the case in other organisms.."
Answer: The word "with" has been replaced by "in" (line13).
5. Line13/14: replace "This review is aimed to provide..." with "This review aims to provide..."
Answer: The words "is aimed to" have been replaced by "aims to" (line 13/14).
6. Line17: replace "about" with "with" and remove comma after hypomorphic
Answer: The word "about" has been replaced by "with" and comma has been deleted (line 17).
7. Line18: replace "mutats" by "mutants
Answer: The word "mutats" has been replaced by "mutants" (line 18).
8. Line35: replace "phonotypes" by "phenotypes"
Answer: The word "phonotypes" has been replaced by "phenotypes".
9. Line36: mention that UFO is an Fbox protein for readers who do not directly come from the field
Answer: UFO is mentioned as F-box protein (line 36).
10. Line38: replace "under-expressed" with "downregulated"
Answer: The word "under-expressed" has been replaced by "down-regulated" (line 38).
11. Line43: is "recruiting" the right word to use? what proteins does CSN recruit to the DNA? Or is CSN just been recruited
Answer: CSN itself gets recruited, the word recruiting has been replaced by "getting recruited" (line 41/42)
12. Line 43: replace "repairs" by "repair"
Answer: The word "repairs" has been replaced by "repair" (line 43)
13. Line 43-45: "In Arabidopsis CSN...UV-induced gene expression", the meaning of the sentence is not clear
Answer: The sentence has been modified as "In Arabidopsis CSN also plays an important role in the DNA repair mechanism. csn7 null mutants are tolerant to UV-C treatment due to constitutive expression of UV-induced genes as CSN negatively regulates ribonucleotide reductase activity" (line 43-46).
14. Line 57: replace "CSN not only..." with "CSN does not only..."
Answer: The sentence starting with "CSN not only" has been replaced with " In addition to interacting with SCFTIR1, the CSN also interacts with other CRLs" (line 57/58).
15. Line 63-66: sentence is not clear
Answer: The sentence has been modified as "DELLA proteins interact with PIF3 and prevent the binding of PIF3 to the target gene promoters. In the presence of GA, DELLA proteins are degraded releasing the PIF3 from the negative suppression of DELLA protein" (line 63-65).
16. Line 70: C-terminal
Answer: C terminal has been replaced with "C-terminal" (line 69)
17. Line 86: write full names of M. sexta and B.cinerea as they are mentioned the first time here
Answer: Full name as "Manduca Sexta and Botrytis cinerea" has been mentioned (line 85)
18. Line 89: replace "Botrytis cinerea" by "B.cinerea"
Answer: "Botrytis cinerea" has been replaced by "B.cinerea" (line 88).
19. Line 96: be consistened with the words rubylated and neddylated, replace rubylated with neddylated
Answer: The word "rubylated" has been replaced with "neddylated" (line 95, 96).
20. Line 113: Where all subunits of the COP9 signalosome identified in this study or just some?
Answer: Hypomorphic mutants for all the subunits of COP9 signalosome is not being identified yet. This study covers all the available hypomorphic mutants.
21. Line126/127: increase the spacing between the lines
Answer: Spacing has been increased between the lines (line 125/126).
22. Line 150: If I understood it right the csn3-3 mutant has a single amino acid exchange from glycine to glutamate. Csn3 can maybe still exert its function similar to the wild type Csn3 and therefore there is also no deneddylation phenotype. Maybe one should be a bit careful with the conclusion that Csn3 has other roles (Line158).
Answer: Hypomorphic mutants such as csn1-10, csn2-5 and csn4-2035 also has only single amino acid exchange but they are defective in Aux/IAA protein degradation as well as in deneddylation activity. Although csn3-3 does not show defect in deneddylation and AUX/IAA degradation but it is defective in auxin response. Thus author (Huang et al., 2013) who produced this line concluded that CSN3 has other role in auxin signaling together with the role of regulating SCFTIR1/AFB ubiquitin ligase. The reference has been incorporated after the completion of this sentence in the manuscript.
23. Line 161: change to "...position 2592..:"
Answer: The change "position 2592" has been made (line 160).
24. Line 164: 2,4D, maybe the authors can write the full name
Answer: Full name of 2,4D "2,4-Dichlorophenoxyacetic acid" has been incorporated (line 163).
25. Line 185: space is missing
Answer: Space has been provided (line 184).
26. Line 186/187: Change the sentence to: "The proteins encoded by these two genes show 87% identity."
Answer: The sentence has been changed to "The proteins encoded by these two genes show 87% identity" (line 185/186).
27. Line 194,195: what about Csn5? Even if there was no effect observed the authors can maybe mention that as this would be also interesting for the reader.
Answer: A sentence " CSN2 and CSN5 are not affected by the loss of CSN6." has been incorporated (line 194).
28. Line 195: give informationabout what is reduced like "...reduces the proteins levels of CSN1,..."
Answer: The words "Protein levels of" have been incorporated (line 193).
29. Line 198: Change to "CUL3 is significantly reduced ..:"
Answer: The suggested change has been incorporated (line 198).
30. Line 201: write "...CUL3 regulate each other reciprocally in such a manner/way that loss of ..:"
Answer: The changes has been incorporated as "CSN5 and CUL3 regulate each other reciprocally in such a manner that loss of either" (line 200, 201)
31. Line 217: "..of both the genes.." replace by "..of both genes"
Answer: The word "the" has been deleted (line 217).
32. Line 217: Authors write that the double deletion of csn6a-1 and csn6b-1 leads to seedling lethality, but in lines 194/195 they state that in the double deletion levels of other CSN subunits are reduced? How does this fit together?
Answer: Double deletion of csn6a-1 and csn6b-1 reduces the level of CSN subunits thus the overall activity of CSN decreases drastically resulting in seedling lethality.
33. Lines 222-226: many repetitions of "adventitious", maybe the authors can replace some of these words
Answer: Adventitious is a correct term to differentiate it from lateral root, any other word may not be correct and can create confusion.
34. Line 228: replace "differently" with "differentially"
Answer: The word "differently" has been replaced with "differentially" (line 228).
35. Line 239: replace "stratification" with "stress"
Answer: In this case, "Stratification" is the more exact term for our conditions. seeds sowing, the plates containing the seeds are kept in cold condition for few days and then moved to the plants growth chamber for uniform germination. This condition is not a stress.
36. Line 255/256: CSN5/1/5A should not be italic
Answer: csn5a-1, and csn5a-2 are hypomorphic mutants and it should be written in italics
37. Line 279: Table1: in den column of deneddylation activity: either "affects" or "does not affect" if the authors want to describe the deneddylation activity with a verb, or "effect" or "no effect" when authors want to state the activity with a noun
Answer: Table 1 column of deneddylation activity has been uniformly maintained with verb "affects" or "does not affect".
38. Line 282: replace "CSN perform" by "CSN performs"
Answer: "CSN perform" has been replaced by "CSN performs (line 282)
39. Line 284: replace "CRLS" by "CRLs"
Answer: The word "CRLS" has been replaces by "CRLs (line 284).
40. Line 297: replace "CSN subunits" by "CSN subunit"
Answer: "CSN subunits" has been replaced by "CSN subunit" (line 297).